# DORA-compliant measures of research quality and impact to assess the performance of researchers in biomedical institutions: Review of published research, international best practice and Delphi survey

**Anna R. Gagliardi**[1]*, **Rob H. C. Chen**[2], **Himani Boury**[1], **Mathieu Albert**[3], **James Chow**[4], **Ralph S. DaCosta**[5], **Michael Hoffman**[5], **Behrang Keshavarz**[6], **Pia Kontos**[6], **Jenny Liu**[7], **Mary Pat McAndrews**[7], **Stephanie Protze**[8]

1 Toronto General Hospital Research Institute, University Health Network, Toronto, Ontario, Canada, 2 UHN Research Solutions and Services, University Health Network, Toronto, Ontario, Canada, 3 The Institute for Education Research, University Health Network, Toronto, Ontario, Canada, 4 Techna Institute, University Health Network, Toronto, Ontario, Canada, 5 Princess Margaret Cancer Centre, University Health Network, Toronto, Ontario, Canada, 6 Toronto Rehabilitation Institute (KITE), University Health Network, Toronto, Ontario, Canada, 7 Krembil Research Institute, University Health Network, Toronto, Ontario, Canada, 8 McEwen Stem Cell Institute, University Health Network, Toronto, Ontario, Canada

* anna.gagliardi@uhnresearch.ca

## Abstract

### Objective

The San Francisco Declaration on Research Assessment (DORA) advocates for assessing biomedical research quality and impact, yet academic organizations continue to employ traditional measures such as Journal Impact Factor. We aimed to identify and prioritize measures for assessing research quality and impact.

### Methods

We conducted a review of published and grey literature to identify measures of research quality and impact, which we included in an online survey. We assembled a panel of researchers and research leaders, and conducted a two-round Delphi survey to prioritize measures rated as high (rated 6 or 7 by ≥ 80% of respondents) or moderate (rated 6 or 7 by ≥ 50% of respondents) importance.

### Results

We identified 50 measures organized in 8 domains: relevance of the research program, challenges to research program, or productivity, team/open science, funding, innovations, publications, other dissemination, and impact. Rating of measures by 44 panelists (60%) in Round One and 24 (55%) in Round Two of a Delphi survey resulted in consensus on the high importance of 5 measures: research advances existing knowledge, research plan is innovative, an independent body of research (or fundamental role) supported by peer-

**Data Availability Statement:** All relevant data are within the paper and its Supporting Information files.

**Funding:** The author(s) received no specific funding for this work.

**Competing interests:** The authors have declared that no competing interests exist.

reviewed research funding, research outputs relevant to discipline, and quality of the content of publications. Five measures achieved consensus on moderate importance: challenges to research productivity, potential to improve health or healthcare, team science, collaboration, and recognition by professional societies or academic bodies. There was high congruence between researchers and research leaders across disciplines.

## Conclusions

Our work contributes to the field by identifying 10 DORA-compliant measures of research quality and impact, a more comprehensive and explicit set of measures than prior efforts. Research is needed to identify strategies to overcome barriers of use of DORA-compliant measures, and to "de-implement" traditional measures that do not uphold DORA principles yet are still in use.

## Introduction

The San Francisco Declaration on Research Assessment (DORA) was established in 2012 during the Annual Meeting of the American Society for Cell Biology [1]. DORA principles advocate for reforming scientific research assessment based on a broad range of discipline-relevant measures of quality and impact. A major tenet of DORA is the elimination of journal-based metrics such as Journal Impact Factor. DORA recommends that academic organizations should be explicit about criteria used for hiring, annual review, tenure, and promotion decisions; assess the value and impact of all research outputs in addition to research publications; and consider a broad range of measures including qualitative indicators of research impact such as influence on policy and practice.

Research norms and outputs vary widely by discipline. Furthermore, journal publications represent only one way to disseminate research, and metrics such as Journal Impact Factor are skewed across disciplines. Therefore, reliance on such metrics is not an accurate, comprehensive, or equitable way to judge the merits of a researcher, or research activity and outputs. For example, qualitative research was rarely published in high-impact general medical and health services research journals over a ten-year period [2]. Also, some journals are not assigned an impact factor although they are peer-reviewed and listed in major research indices; thus, reliance on Journal Impact Factor risks overlooking high-quality research published outside of journals considered "high impact". The harms associated with evaluating research based on Journal Impact Factor and questions about the validity of how the impact score is determined have long been recognized [3–8].

While the value of interdisciplinary or team science is widely recognized [9], research shows that many researchers struggle to achieve legitimacy in biomedical settings [10]; for example, qualitative health care researchers, whose political and epistemological orientation and research processes are in opposition to positivism [11]. Some have noted that continued reliance on the "latent biomedical conservatism that characterizes the health sciences" combined with a lack of frameworks that acknowledge and properly assess diverse forms of scholarship disadvantage many researchers and impede their professional advancement [12].

Given the many deficiencies of journal metrics for assessing research productivity and contributions, it is no wonder that the DORA principles have been widely endorsed. As of March 23, 2022, 21,385 individuals and organizations in 156 countries are DORA signatories.

However, a challenge to implementing DORA principles is the lack of established alternatives to journal metrics. For example, the Leiden Manifesto offers 10 principles that uphold DORA principles (e.g. measure performance against research institute mission, account for variation by field in publication and citation practices) [13]. A 2017 meeting of international experts in scientific communication generated five principles upon which to judge research: assess contribution to societal needs, employ responsible indicators, reward publishing of all research regardless of the results, recognize the culture of open research, fund research that generates evidence on optimal ways to assess science and faculty, and fund/recognize out-of-the-box ideas [14]. While helpful in terms of guidance and advocacy, these principles may not represent a comprehensive list of measures for assessing research quality and impact. Others have suggested criteria for research assessment, but they are discipline-specific and not broadly applicable to diverse fields of research. For example, Mazumdar et al. proposed measures to assess the contributions of biostatisticians to team science [15].

As a DORA signatory, our organization formed a DORA advisory group (authors of this manuscript) representing different research disciplines and stages of career to align research reporting and assessment with DORA principles. To achieve this, we aimed to identify and prioritize measures for assessing the performance of researchers that comply with DORA principles by focusing on quality and impact rather than metrics such as Journal Impact Factor. We used an evidence- and consensus-based approach that generated 10 measures and identified processes to support uptake of those measures within our institution. Researchers in our organization and elsewhere can employ these measures to describe and promote the value of their research, and academic organizations can employ these measures and related processes to assess the performance of researchers in the context of hiring, annual reviews, tenure, promotion and other decisions based on the quality and impact of research. The purpose of this manuscript is to describe our methods and the resulting DORA-compliant measures for assessing the performance of researchers.

## Methods

### Research design

We conducted a sequential, multi-methods study. We assembled research assessment measures and processes by conducting a scoping review of published and grey literature [16]. We chose a scoping review over other types of syntheses because it is characterized by the inclusion of a range of study designs, which facilitates the exploration of literature in a given field and reveals the nature of existing knowledge [17,18]. Similar in rigor to a systematic review, a scoping review does not assess the methodological quality of included studies and does not assume or generate a theoretical stance. We supplemented published research by searching "grey" literature, referring to a range of types of documents (e.g. academic publications, strategic plans, program evaluations) available on the Internet. Grey literature searching is challenging because there are few dedicated repositories of grey literature, no standard methods for searching for grey literature, and the effort required is inversely related to the typically low yield [19,20]. However, we chose to do so for this study as the DORA website provides links to reports of international initiatives that adopted and applied DORA principles. Scoping review findings formed the basis of a Delphi survey. The Delphi technique is a widely used method for generating consensus on strategies, recommendations, or quality measures [21–23]. This technique is based on one or more rounds of survey in which panelists independently rate recommendations until a degree of consensus is achieved. We did not register a protocol. We consulted with the University Health Network Research Ethics Board, who determined that

we did not require ethics approval for this initiative. We complied with research reporting criteria for scoping reviews [24] and Delphi studies [25].

## Scoping review

**Eligibility.** Author ARG conducted a preliminary search in MEDLINE using the Medical Subject Headings "employee performance appraisal" AND "research personnel" to become familiar with the literature, draft eligibility criteria, and inform a more comprehensive search strategy. All authors reviewed and refined PICO-based eligibility criteria, which applied to both the database search and the grey literature search. S1 Appendix details inclusion and exclusion criteria. In brief, we included studies in which participants were researchers from multiple research disciplines or research leaders based in academic settings. Research disciplines reflected the *Canadian Research and Development Classification 2019* developed by the Tri-Council Funding Agencies [26]. The issue referred to research productivity, contributions to science, health systems or society, quality or impact, or other synonymous terms used by eligible studies. Comparisons, or the purpose of assessment included hiring, annual review, reappointment, compensation, tenure, promotion, consideration for leadership or other awards, etc. With respect to publication type, we included any qualitative, quantitative, or multiple-/mixed-methods study. Outcomes included measures, indicators, criteria, suggestions, recommendations, policies, or practices for research assessment; or the preferences of researchers or research institutes for measures or processes related to research assessment. We did not include measures reflecting the assessment of trainees or trainee research, non-research measures (e.g. teaching, supervision, other services), or diversity due to a concurrent effort underway at UHN with a focus on equity.

**Searching and screening.** ARG, who has medical librarian training, developed and executed searches (S2 Appendix), complying with search strategy reporting guidelines [27]. We searched MEDLINE, EMBASE, CINAHL, AMED, and the Web of Science for studies published in English language from 2013 to January 15, 2021. We chose 2013 because DORA principles were published in 2012, following which one might expect publications for research based on DORA principles. All search results were imported into Covidence to remove duplicates and facilitate screening. Authors ARG, BK, and MA independently screened the first 20 titles and abstracts and compared and discussed findings. This identified only one discrepancy in the selection of eligible studies, which was resolved by further refining eligibility criteria. Thereafter, ARG screened the remaining titles and abstracts and acquired full-text versions of potentially-eligible articles.

**Data collection and analysis.** ARG extracted data from eligible articles on first author, year of publication, country, study objective, research design and key findings (i.e. research assessment measures or related processes). Based on that data, ARG compiled a list of unique measures and related processes that could be implemented to promote awareness and support use of the measures reported or recommended across all included studies.

**Grey literature.** We employed a targeted approach to search for publicly available grey literature by browsing the DORA website (https://sfdora.org/), following links from the DORA website to international organizations, and both browsing and searching the websites of Canadian universities. On each site, we searched for institutional policies, strategic plans, or other documents that described research assessment measures, or reporting or evaluation processes. ARG searched for relevant reports that met eligibility criteria (S1 Appendix) and extracted data on: organization name, title of the document or website, year published, document purpose, and research assessment measures or processes. ARG compiled a list of unique measures and processes, and integrated those with the list of measures and processes compiled from

published research, resulting in 50 unique measures organized in eight categories that inductively emerged: relevance of research program, challenges to productivity, team/open science, funding, innovations, publications, other dissemination, and evidence of impact. First, ARG perused all measures using content analysis to organize measures of similar theme in categories reflecting those themes; for example, measures that explicitly mentioned or pertained to team science, participation, co-production, registration or sharing of research or open publication were categorized under Team/Open Science; and measures that explicitly mentioned or pertained to peer-reviewed research funding, applications funded or grants were categorized under Funding. Members of the research team independently reviewed, discussed and agreed upon the categorization scheme.

## Delphi survey

**Survey development.**   All authors reviewed the integrated list of measures, processes and refined wording. These measures formed the basis of the Delphi survey that was administered using an in-house application that creates online surveys.

**Sampling and recruitment.**   A review of Delphi studies showed that the median number of panelists was 17 (range from 3 to 418) [23]. Other research found that the reliability of Delphi rating increased with panel size [22]. To ensure that panelists represented multiple perspectives, we aimed to include persons who varied by: research institute within our organization, research discipline, career stage (early, mid, late), and professional role: researcher or research leader (e.g. head of the research institute). To compile the list, we referred to institutional databases and asked research institute administrators for suggestions. This resulted in a 74-member panel of researchers, of which 6 (8.1%) were in leadership positions. By research institute focus, this included: 18 rehabilitation science, 10 brain-related conditions research, 1 stem cell research, 20 cancer research, 3 research on technology in health, 12 multidisciplinary biomedical and health systems research in cardiovascular, endocrine, infectious, kidney, liver and lung diseases; and 10 healthcare education. By type of research, this included 31 biomedical, 21 clinical, 11 health services and 11 social/cultural researchers. Of the 74 panelists, 24 (32.4%) were early career, 28 (37.8%) midcareer and 22 (29.7%) late-career, referring to < 5 years, 5–10 years and > 10 years as independent researchers, respectively. Due to reasons of privacy and confidentiality, we did not have access to data on gender, age or ethno-cultural characteristics of panelists, nor did we report the research institute of individual participants, which combined with specified role, could identify individuals.

**Data collection and analysis.**   We asked panelists to rate each recommendation on a 7-point Likert scale (1 = strongly disagree, 4 = neutral, 7 = strongly agree), comment on the relevance or wording of each recommendation if desired, and suggest additional recommendations not included in the survey. Standard Delphi protocol suggests that two rounds of rating with agreement by at least two-thirds of panelists to either retain or discard items will prevent respondent fatigue and drop-out [20–22]. We followed these suggestions and conducted two rounds of rating. We emailed Instructions and a Round One survey link to panelists on July 21, 2021, with reminders at one and two weeks following the initial invitation. Based on the results, we developed a Round One summary report that included Likert scale response frequencies and comments for each recommendation, which we organized by those retained (rated by at least 80% of panelists as 6 or 7), discarded (rated by at least 80% of panelists as 1 or 2) or no consensus (all others), along with newly suggested recommendations. On September 15, 2021, we emailed panelists the Round One summary report with a link to the Round Two survey, formatted similarly to the Round One survey, to prompt rating of recommendations that did not achieve consensus for inclusion or exclusion in Round One. We

emailed a reminder at one, two and three weeks after the initial invitation. We analyzed and summarized Round Two responses as described for Round One. Ultimately, because few measures were highly rated by 80% of panelists, we retained measures that achieved high (rated 6 or 7 by ≥ 80% of panelists) or moderate (rated 6 or 7 by ≥ 50% of panelists) consensus.

## Results

### Scoping review

Of 1,566 unique search results, we excluded 1,538 titles, and of 28 potentially eligible full-text articles, we excluded 17 due to a focus on publication metrics (9), ineligible publication type (6), no research assessment measures or processes reported (1), and context was not biomedical (1). Ultimately, 11 articles were included for review (Fig 1). S3 Appendix includes data extracted from eligible articles [28–37]. Of 55 grey documents (27 from Canadian university websites; 28 from DORA and other international organizations), we excluded 29 because they did not contain relevant content and included 26 documents in the review. S4 Appendix includes data extracted from the eligible documents [38–62].

### Compiled measures and processes

S5 Appendix shows the list of the 50 unique research assessment measures compiled from published and grey literature. S6 Appendix shows the list of unique processes that organizations can implement to promote awareness and support use of those measures. Table 1 includes select illustrative examples of those processes, which create a culture conducive to DORA-compliant assessment of research performance based on research quality and impact.

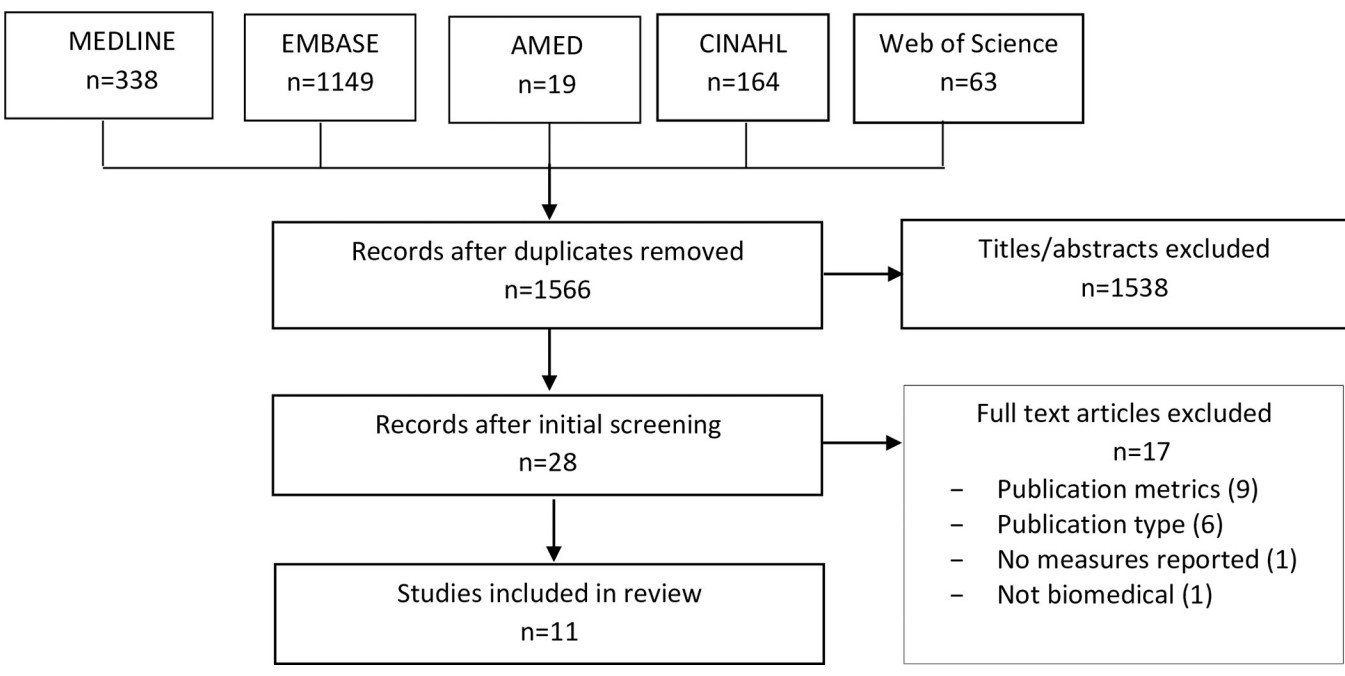

**Fig 1. PRISMA diagram of articles included in review of published research.**

**Table 1. Select processes to support uptake of DORA-compliant measures.**

| Category | Processes |
|---|---|
| Principles | • Create a more porous research culture to promote interdisciplinary approaches, and enable more mobile and flexible research careers<br>• Have individuals highlight and articulate their most meaningful contributions by describing the quality, significance and impact of their scholarship, and specify the level of impact: individual, community, system and population to help reviewers assess its merits |
| Responsibilities | • Research managers and administrators should champion these principles and the use of responsible evaluation within their institutions<br>• Individual researchers are responsible for providing narrative descriptions of the quality, context, and impact of their research, individual research products, and other aspects of research without relying on surrogate metrics. The person best placed to articulate the importance of the research is the researcher themselves. |
| Review processes | • Assemble diverse review committees reflecting inclusivity, diversity, equity and ability—across gender, seniority, cultures, and under-represented minority populations—to bring a range of perspectives and experiences into decisions<br>• To facilitate success, each Research Institute or Unit Chair shall: Review with each member their responsibilities and expectations; Meet with each member annually to discuss their annual report; and performance quality, progress and trajectory; Discuss career goals, and offer mentorship and other supports; Discuss merit recommendations and jointly agree on an action plan to address deficiencies |
| Implementation of DORA-compliant measures | • Provide education/training to research leaders and researchers about these principles, and how to assess (leaders) and describe (researchers) research productivity/contributions<br>• Incentivize and reward a broader range of academic activities |

## Delphi survey

**Panelists.** Of 74 researchers invited to participate, 44 (59.5%) completed the Round One survey. Of those 44, 24 (54.5%) completed the Round Two survey. By research institute focus, his included 4 rehabilitation science, 3 brain-related conditions research, 9 cancer research, 2 research on technology in healthcare, 2 multidisciplinary biomedical or health systems research, and 2 healthcare education. By type of research, this included 12 biomedical, 5 clinical, 4 health services and 3 social/cultural. Overall, 4 (66.7%) invited research leaders and another 40 (58.8%) invited researchers participated in at least one survey round (Table 2).

**Delphi rating.** S7 Appendix shows respondent ratings of all measures and S8 Appendix lists the pros and cons offered by respondents for all measures that did not achieve high or moderate consensus. Fig 2 summarizes results across two rounds of rating.

**Table 2. Delphi respondent characteristics.**

| Respondents | Survey Round respondents/recipients (%) | |
|---|---|---|
| | **One** | **Two** |
| **Role** | | |
| Researchers | 40/68 (58.8) | 22/40 (55.0) |
| Researchers in leadership role | 4/6 (66.7) | 2/4 (50.0) |
| **Researchers** | | |
| Early career | 11/24 (45.8) | 5/11 (45.5) |
| Mid career | 19/28 (67.9) | 9/19 (47.4) |
| Late career | 14/22 (63.6) | 10/14 (71.4) |
| Total | 44/74 (59.5) | 24/44 (54.5) |

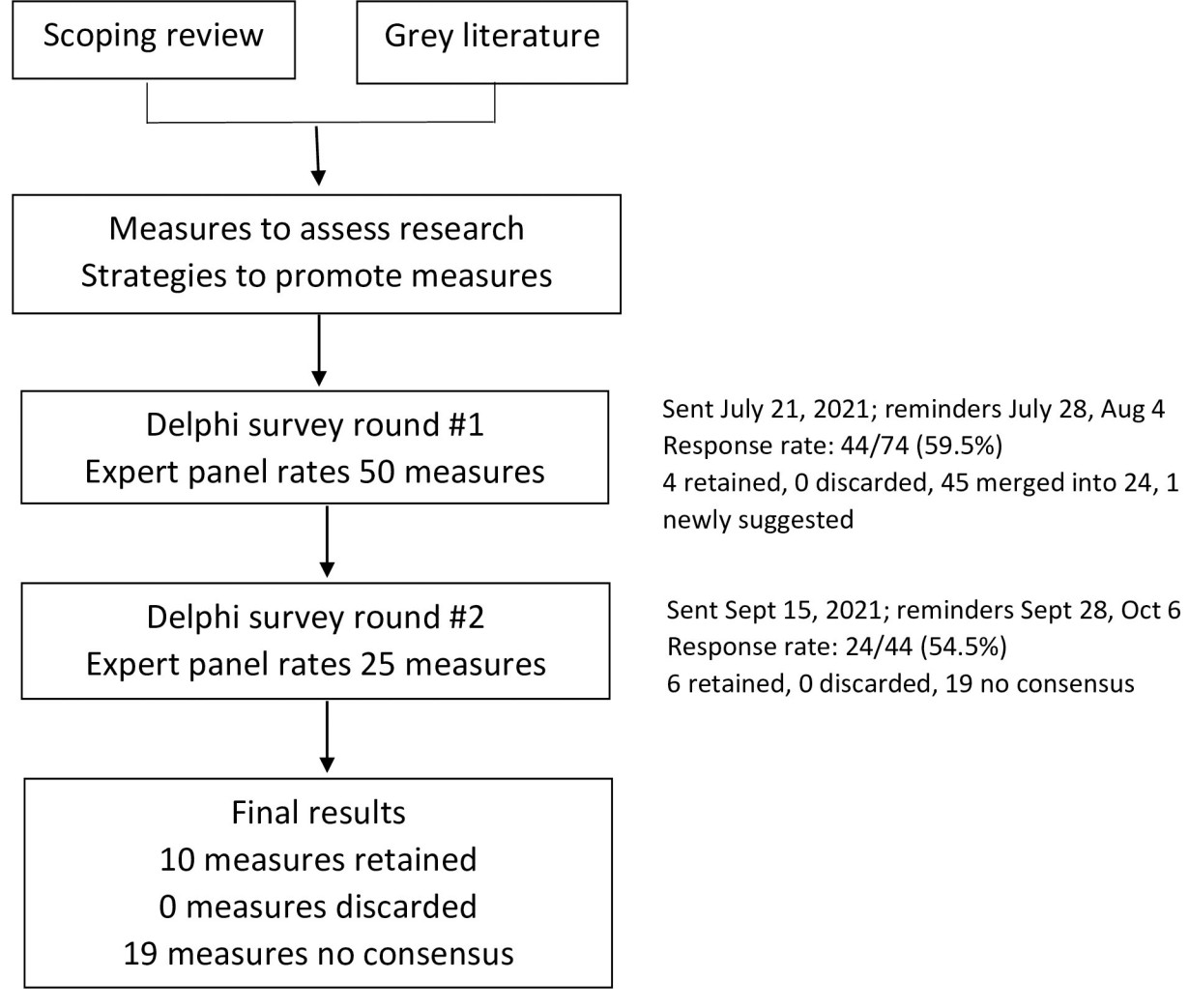

**Fig 2. Delphi process and results.**

**Recommended measures.** Ten measures achieved high or moderate consensus (Table 3).

## Discussion

Academic organizations assess research activity and outputs, yet meaningful measures for doing so are lacking. In this study, we reviewed published research and grey literature to derive measures of research quality and impact that could be used to assess researcher activity and impact organized in 8 domains: relevance of research program, challenges to research program or productivity, team/open science, funding, innovations, publications, other dissemination, and impact. A two-round Delphi survey resulted in consensus on 10 measures, including 5 measures of high importance: research advances existing knowledge, the research plan is innovative, an independent body of research (or fundamental role) supported by peer-reviewed research funding, research outputs relevant to the discipline, and quality of the content of publications; and 5 measures of moderate importance: challenges to research productivity, potential to improve health or healthcare, team science, collaboration, and dissemination or recognition by professional societies or academic bodies. There was high congruence between

**Table 3. Measures that achieved consensus to retain.**

| Category | Measure (degree of consensus) |
|---|---|
| Relevance of research program | 1. Research advances existing applied and/or theoretical knowledge (high)<br>2. Research plan is innovative (e.g. generates novel methods, models, data, or other knowledge that addresses a noted gap) (high)<br>3. Research directly addresses or has the potential to improve healthcare and the health of the public (moderate) |
| Funding | 4. Evidence of research independence (e.g. PI or co-PI) OR of a fundamental role on a research team (e.g. biostatistician, qualitative researcher) with peer-reviewed research funding (high) |
| Innovations | 5. Evidence of research outputs/products relevant to type of research (researcher can choose from the following list or provide other relevant options): (high)<br>• Commercialization of technology (e.g. software, drugs, devices), launch of companies, invention disclosures, patent applications, issued patents<br>• Creation of cohorts or registries<br>• Reusable software or datasets<br>• Reusable reagents (plasmids, mouse models, cell lines)<br>• Clinical tests, algorithms or statistical models<br>• Validated questionnaires or instruments<br>• Contribution to policies, standards, guidelines or programs<br>• Novel theory, model or framework<br>• Novel research approaches, or methods<br>• Other forms of achievement or outputs relevant to discipline (e.g. performance art) |
| Publications | 6. Quality of the content of publications as judged by a peer reviewer or panel, in part based on rationale provided by the researcher of importance to their field (high) |
| Team Science | 7. Where it would benefit the research topic or program, evidence of local, regional, national or international team science (e.g. formation of clinical network or inter-/multi-disciplinary research group) (moderate)<br>8. Evidence of collaboration or multidisciplinary research through participation as a co-investigator or mentor on other research teams (moderate) |
| Recognition | 9. Recognition by academic or professional societies (e.g. awards, honours) (moderate) |
| Challenges to research productivity | 10. Qualitative description by researcher of challenges faced and mitigating strategies applied (e.g. leaves of absence for family or medical reasons) (moderate) |

respondents with different roles (e.g. researcher, research leader) sampled to represent a wide range of research disciplines.

Prior research that examined research assessment found that it relied on largely on journal metrics and revealed few frameworks of assessment measures with little agreement across frameworks. For example, a 2018 and survey of criteria used for assessing researchers at 92 international faculties of biomedical sciences revealed they largely employed traditional measures such as the number of peer-reviewed publications, impact factor, and number or amount of grant funding [22]. A 2018 scan of Canadian faculty of health sciences and medicine websites identified few frameworks used to support hiring or promotion, and those identified employed vague statements about creativity or quality but no explicit measures [12]. In a survey of medicine and life sciences faculty at five Belgian universities, 126 respondents rated publishing in high impact journals or publishing more papers than others as contributing more to advancing careers rather than advancing science or personal satisfaction, and rated having research results used or implemented higher on both scientific advancement and personal satisfaction compared with career advancement [63]. A 2021 editorial on research impact stated there are more than 20 frameworks to understand and evaluate research impact, but noted they are context-specific, vary widely in the outcomes they emphasize and lack empirical validation [64]. These studies underscore the lack of explicit non-metric based measures for assessing research, a gap that our research addressed.

Our work builds on a 2017 meeting of 22 experts from the United States, England, Germany, Netherlands and Canada who reviewed select literature critiquing traditional research assessment and generated five principles upon which to judge research: societal benefit, contributions to science, out-of-the-box ideas, full and transparent publication regardless of results and open science [14]. Our work generated measures that match these first 3 principles plus an additional 7 measures by which to assess research activity and outputs, as recommended by DORA. Clearly, there is a paucity of research on non-traditional measures for assessing research given that we identified only 11 empirical studies on this topic published after the release of the DORA principles in 2012 [1]. Given a lack of insight on a range of relevant measures for assessing research, our work contributes to the field by generating consensus on non-traditional measures of research activity, quality, and impact that can be used to uphold DORA principles in our organization and other academic organizations worldwide who already endorsed DORA or are contemplating how to do so.

The 10 measures generated by this research can be used by researchers when reporting on the quality and impact of their research as part of performance assessment, and by employers or evaluators when assessing the performance of researchers for hiring, annual review, tenure, promotion and other decisions. Academic research organizations and others (e.g. funders) can compare their research evaluation rubrics and processes to the measures identified by this study as a means of planning or enhancing the way that the performance of researchers is assessed. Of further support are the processes, including principles, responsibilities and approaches by which to apply these measures, and promote awareness, adoption and use of the measures on the part of researchers and employers/evaluators. This work will be directly relevant to the 22,311 individuals and organizations in 159 countries who have officially endorsed DORA (as of November 1, 2022).

In a broader context, these findings are also germane to discussions about the tangible value of research. Governments and funders worldwide are placing increasing emphasis on the assessment of research impact to supply evidence of the value of their research investments to society [65]. To foster research impact, national-level initiatives in the United Kingdom (Collaborations for Leadership in Applied Health Research and Care) and the Netherlands (Academic Collaborative Centres) invested heavily in implementing regional networks of researchers or academic organizations, government policymakers, health system leaders, and members of the public or representing healthcare advocacy groups [66,67]. These networks are based on the concepts of participatory research or integrated knowledge translation, whereby research is more likely to be relevant and used when planned from the outset with target users [68]. Evaluations of these entities revealed they improved service delivery and associated clinical outcomes [69]. In 2014, the United Kingdom established the Research Excellence Framework, which defined research impact as: "an effect on, change or benefit to the economy, society, culture, public policy or services, health, the environment or quality of life, beyond academia." [70]. The Framework was accompanied by over 6,000 case studies demonstrating research impact. Analysis of a subset of high-impact case studies revealed the most common forms of impact: practice (e.g. changing professional behaviour, and improving organizational culture, quality of services, and outcomes), government policy (e.g. adopting new policies, reducing costs), economic (e.g. greater revenue, profit or market share) and public awareness (e.g. improving public knowledge or attention to an issue) [71]. UK Research and Innovation is currently (as of December 2021) introducing a new Resume for Research and Innovation for evaluating scientists that relies heavily on context instead of raw metrics (https://www.ukri.org/news/ukri-launches-new-resume-for-research-and-innovation/I). In addition to initiatives like DORA [1], such national-level efforts that value research based on the good it can achieve may well contribute to a declining reliance on journal metrics.

However, in this study, measures reflecting co-production of research with those outside of academia; research reflecting the needs and preferences by sex, gender and intersectional factors; and evidence of societal research impact were rated of low importance. Furthermore, such practices are increasingly required by funders of health services research and related disciplines, and if not included in assessment rubrics, may result in health services researchers being held far more accountable than other disciplines for resource- and time-intensive activities that go unrecognized by employers or assessors who continue to rely on traditional research metrics. The adoption of new practices can be slow, particularly when the necessary change requires a profound culture shift, as is the case with DORA principles of research assessment. Thus, additional research is needed to understand the perceived and actual value of DORA principles and our measures, barriers to their uptake, and the knowledge and strategies needed to address these barriers. This will be critical to informing interventions that support the embracing of new measures of research quality and impact, and to "de-implement" traditional measures that are inconsistent with DORA and deemed inappropriate, yet are still in use. One way to do this is to learn how other organizations who have successfully adopted DORA-compliant measures and processes achieved the culture shift. Recognizing that culture shift may be a major barrier to adopting the measures recommended in this report, ongoing research is needed to assess the perceived value of these measures and barriers to their uptake, knowledge needed to select and tailor strategies or interventions aimed at supporting uptake. For example, measures must be reported qualitatively by researchers, and judged qualitatively by those with expertise in a relevant discipline, which can be more involved and time-consuming than traditional quantitative metrics such as counting number of publications. Because changes in research assessment may have broader implications, interviews should also be conducted with non-researcher staff such as human resources, or managers responsible for compiling and analyzing annual or periodic research activity reports submitted by researchers. Also, forging strategic and tactical alliances with academic organizations, publishers, and funding bodies will be necessary to achieve the successful uptake of non-traditional measures.

This study features several strengths. The measures rated by panelists were derived from research and international best practices. We assembled a panel comprised of researchers representing different research roles and disciplines. The large panel size enhanced reliability. Two rounds of rating minimized respondent fatigue, which achieved a high response rate in both rounds. We optimized rigor by complying with methodology and reporting criteria for scoping reviews and Delphi studies [24,25]. Findings are bolstered by the high congruence in rating between researchers, research leaders, and those representing different research disciplines and career stages. We must also acknowledge some limitations. Our search for sources of measures may not have been sufficiently comprehensive and only 11 papers on the subject published since 2012 emerged; however, as part of the Delphi process, panelists were asked to identify additional measures not already included in the survey. For reasons of privacy and confidentiality, we did not have access to respondents' personal details, and therefore could not examine ratings of measures by gender, age or ethno-cultural characteristics; however, ratings were congruent, so sub-analyses may not have yielded meaningful differences. Respondents' views may differ from those of researchers or research leaders in other jurisdictions. The findings may not be generalizable in countries outside of Canada with differing scientific or academic cultures and structures. However, numerous organizations worldwide have embraced DORA, so the measures generated in our work are likely relevant at organizational level.

In conclusion, a two-round Delphi survey of researchers and research leaders representing a range of scientific disciplines, based on compilation of measures of research assessment from published and grey literature, resulted in consensus on ten measures compliant with DORA

principles that can be used by researchers to report on the quality and impact of their research activity, and by employers/evaluators to assess researchers for performance evaluation in the context of hiring, annual review and promotion or tenure decisions.

## Supporting information

**S1 Appendix. Eligibility criteria for published research.**
(DOCX)

**S2 Appendix. Search strategy for published research.**
(DOCX)

**S3 Appendix. Data extracted from included articles.**
(DOCX)

**S4 Appendix. Eligible documents identified in grey literature.**
(DOCX)

**S5 Appendix. Research assessment measures compiled from published and grey literature.**
(DOCX)

**S6 Appendix. Processes to support uptake of research assessment measures.**
(DOCX)

**S7 Appendix. Respondent ratings of all measures.**
(DOCX)

**S8 Appendix. Pros and cons reported by respondents for measures not prioritized.**
(DOCX)

## Acknowledgments

We acknowledge Joan Wither and Catriona Steele, who contributed to early-stage decision-making, and Stephanie Susman, who assisted with data collection.

## Author Contributions

**Conceptualization:** Anna R. Gagliardi, Rob H. C. Chen, Mathieu Albert, James Chow, Ralph S. DaCosta, Michael Hoffman, Behrang Keshavarz, Pia Kontos, Jenny Liu, Mary Pat McAndrews, Stephanie Protze.

**Data curation:** Anna R. Gagliardi, Rob H. C. Chen, Himani Boury, Mathieu Albert, James Chow, Ralph S. DaCosta, Michael Hoffman, Behrang Keshavarz, Pia Kontos, Jenny Liu, Mary Pat McAndrews, Stephanie Protze.

**Formal analysis:** Anna R. Gagliardi, Rob H. C. Chen, Mathieu Albert, James Chow, Ralph S. DaCosta, Michael Hoffman, Behrang Keshavarz, Pia Kontos, Jenny Liu, Mary Pat McAndrews, Stephanie Protze.

**Funding acquisition:** Anna R. Gagliardi.

**Investigation:** Anna R. Gagliardi, Rob H. C. Chen, Mathieu Albert, James Chow, Ralph S. DaCosta, Michael Hoffman, Behrang Keshavarz, Pia Kontos, Jenny Liu, Mary Pat McAndrews, Stephanie Protze.

**Methodology:** Anna R. Gagliardi, Rob H. C. Chen, Mathieu Albert, James Chow, Ralph S. DaCosta, Michael Hoffman, Behrang Keshavarz, Pia Kontos, Jenny Liu, Mary Pat McAndrews, Stephanie Protze.

**Project administration:** Anna R. Gagliardi, Rob H. C. Chen, Himani Boury.

**Resources:** Anna R. Gagliardi.

**Supervision:** Anna R. Gagliardi.

**Writing – original draft:** Anna R. Gagliardi, Rob H. C. Chen, Himani Boury, Mathieu Albert, James Chow, Ralph S. DaCosta, Michael Hoffman, Behrang Keshavarz, Pia Kontos, Jenny Liu, Mary Pat McAndrews, Stephanie Protze.

**Writing – review & editing:** Anna R. Gagliardi, Rob H. C. Chen, Himani Boury, Mathieu Albert, James Chow, Ralph S. DaCosta, Michael Hoffman, Behrang Keshavarz, Pia Kontos, Jenny Liu, Mary Pat McAndrews, Stephanie Protze.

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
