## [Decision Letter · Decision Letter 0]

27 Oct 2022

PONE-D-22-16889DORA-compliant measures to assess research quality and impact in biomedical institutions: review of published research, international best practice and Delphi surveyPLOS ONE

Dear Dr. Gagliardi,

Thank you for submitting your manuscript to PLOS ONE. After careful consideration, we feel that it has merit but does not fully meet PLOS ONE’s publication criteria as it currently stands. Therefore, we invite you to submit a revised version of the manuscript that addresses the points raised during the review process.

ACADEMIC EDITOR:

Thank you to the authors for submitting this work to PLOS ONE and for their patience during the review process. I apologize for the delay in securing reviewers. It took time to find reviewers with appropriate expertise and many candidates were unavailable over the summer months. 

This is a great scoping review and Delphi panel with interesting and useful results. Table 3 and Appendix 5 are very high value. Research impact assessment is an important undertaking. I hope to ultimately see wide adoption of the measures you have identified.  

Reviewer 2 had only minor comments. 

Reviewer 1 had more substantial comments on a potential discrepancy in focus between the data arising from the database search and the grey literature search. The reviewer suggested heavily revised framing of the manuscript to accommodate the wider focus of the grey literature data. 

I did not consider there to be a conceptual mismatch in the focus of the database data and grey literature data. However, please address the reviewers comments and consider minor reframing of some elements of the manuscript to ensure uniformity in the focus of the work. 

In addition to the reviewers comments I have three suggested revisions:

1.Would it be possible to report the research institutes and/or research domains of the participants? On page 23 line 396 you indicate that the sample included different research disciplines. While you describe sampling from different research backgrounds there is no reporting on the institutes or disciplines of the respondents. Reporting this may strengthen the results as it will show that there was a congruency of ratings across disciplines. This would also dispel potential criticism that the uniformity may have arisen from a homogenous disciplinary background of the participants. If this is not permitted due to the ethical constraints of your data collection, then please indicate so in the manuscript.

2.There may be a typo on page 9 line 191 where you report 49 unique measures. In the rest of the manuscript you report 50 unique measures.  

3.Page 8, Line 149: would consider replacing the wording “a wide array of research disciplines” with “multiple research disciplines” as it’s a more reproducible criterion that does not require further definition.

We look forward to receiving your revised manuscript.

Kind regards,

Niklas Bobrovitz

Academic Editor

PLOS ONE

Journal Requirements:

2. Please amend your authorship list in your manuscript file to include author Himani Boury.

Reviewers' comments:

Reviewer's Responses to Questions

**Comments to the Author**

1. Is the manuscript technically sound, and do the data support the conclusions?

Reviewer #1: Partly

Reviewer #2: Yes

2. Has the statistical analysis been performed appropriately and rigorously? 

Reviewer #1: N/A

Reviewer #2: N/A

3. Have the authors made all data underlying the findings in their manuscript fully available?

Reviewer #1: Yes

Reviewer #2: Yes

4. Is the manuscript presented in an intelligible fashion and written in standard English?

Reviewer #1: Yes

Reviewer #2: Yes

5. Review Comments to the Author

Reviewer #1: Thank you for the opportunity to review the paper. I always aim to provide constructive comments that hope to improve the draft paper. I have utilised the PLOS criteria as the basis for review:

1. The study presents the results of original research.

Yes. I don't think the research is particularly novel, but the identification of domains from the specified PICO criteria and the Delphi review of the identified domains is original and consequently, contributes original research evidence.

2. Results reported have not been published elsewhere.

Not for this study, of which I am aware.

3. Experiments, statistics, and other analyses are performed to a high technical standard and are described in sufficient detail.

The scoping review may be sufficient for this form of research and appears to have been conducted in line with PRISMA-Scr guidelines. It would have been preferable for the grey literature to be assessed using the PICO developed for the academic database search. The grey literature contained a higher number of the relevant publications and yet, based on my reading of the text, the review process was separate from the specified PICO and accordingly introduced additional subjectivity, did not align with the PICO process/outcomes, and lost the focus on the research quality and impact assessment domains applied to research performance assessment/promotion. The argument could be made that the wider points drawn from the grey literature are pertinent, but if this was the case, the academic literature should have also drawn this wider remit. As a consequence, the scoping review conflates an historical review of the domains applied in the performance review/assessment process and a broader review of what could be incorporated. The latter component was however excluded from the academic database search and consequently misses a large volume of potentially relevant literature. Given the existing summation of potential domains provided the basis for the Delphi process, it is difficult to see how this could be comprehensively adjusted. Greater clarity regarding the inclusion/exclusion criteria for the grey literature may assist. This conflation is also relevant to the following point and the results, discussion and conclusion issue identified below.

The specified aim of the paper is 'to identify and prioritize DORA-compliant measures for assessing research.' This aim is inconsistent with the PICO, which, necessarily in this reviewer's subjective opinion, focuses on 'Personnel Selection/personnel management/employee incentive plans/employee performance appraisal/academic performance/career mobility' i.e. the assessment of researchers within a view to workplace performance appraisal, promotion, etc. I would recommend tightening the aim to reflect this narrower focus and similarly to refine the specified purpose to also reflect the performance context for the relevant measures.

4. Conclusions are presented in an appropriate fashion and are supported by the data.

The paper results, discussion and conclusion needs to be carefully reviewed to improve precision and alignment with the PICO focus on researcher performance appraisal. For example, greater clarity is required to explain how Table 1/S6 Appendix links to the aims (or preferably narrower aims that reflect the PICO strategy).

5. The article is presented in an intelligible fashion and is written in standard English.

Yes

6. The research meets all applicable standards for the ethics of experimentation and research integrity.

Yes

7. The article adheres to appropriate reporting guidelines and community standards for data availability.

Yes

Additional general points:

- As per the critique relating to the drift of focus from research performance assessment/promotion, the title is similarly ambiguous. I would recommend changing the title, aim, purpose, and where possible, the results, discussion and conclusion to this narrower focus. The field of research on research is generally under-researched, the initiative is valuable and the focus on researcher performance assessment is a highly relevant issue.

Reviewer #2: The review and Delphi study aimed to identify and prioritize DORA-compliant measures for assessing research quality and impact. Broadly speaking, this work is important and timely as we need better metrics for determining the quality and impact of research. This work has the potential to contribute greatly to the field.

Overall, this manuscript is very high quality and meets standard reporting guidelines. It is transparent, thorough, and enjoyable to read. It is difficult to find areas for further improvement; however, I suggest two minor revisions:

- Please include search strategy for all databases, the appendix only includes MEDLINE

- Line 192 you discuss how eight categories “inductively emerged.” Please describe in detail the methods you used for this inductive process (e.g. how the data were coded, use of thematic analysis, content analysis, if it was independently in duplicate or collaboratively, etc.)

6. PLOS authors have the option to publish the peer review history of their article (what does this mean?). If published, this will include your full peer review and any attached files.

Reviewer #1: No

Reviewer #2: No

---

## [Author Response · Author response to Decision Letter 0]

1 Nov 2022

EDITOR

1/

Reviewer 1 had more substantial comments on a potential discrepancy in focus between the data arising from the database search and the grey literature search. The reviewer suggested heavily revised framing of the manuscript to accommodate the wider focus of the grey literature data. I did not consider there to be a conceptual mismatch in the focus of the database data and grey literature data. However, please address the reviewers comments and consider minor reframing of some elements of the manuscript to ensure uniformity in the focus of the work.

AUTHORS: Thanks very much for your guidance. Please see below how we addressed Reviewer #1’s concern

2/

Would it be possible to report the research institutes and/or research domains of the participants? On page 23 line 396 you indicate that the sample included different research disciplines. While you describe sampling from different research backgrounds there is no reporting on the institutes or disciplines of the respondents. Reporting this may strengthen the results as it will show that there was a congruency of ratings across disciplines. This would also dispel potential criticism that the uniformity may have arisen from a homogenous disciplinary background of the participants. If this is not permitted due to the ethical constraints of your data collection, then please indicate so in the manuscript.

Authors: On page 10-11, lines 213-215, we note: “…nor did we report the research institute of individual participants, which combined with specified role, could identify individuals.” However, we did report the research institute focus and discipline of the initial panel (page 12, lines 222-227) and the same for the final respondents (page 16, lines 281-285)

3/

There may be a typo on page 9 line 191 where you report 49 unique measures. In the rest of the manuscript you report 50 unique measures. 

Authors: Done

4/

Page 8, Line 149: would consider replacing the wording “a wide array of research disciplines” with “multiple research disciplines” as it’s a more reproducible criterion that does not require further definition.

Authors: Done

REVIEWER #1

1/

The study presents the results of original research.

Yes. I don't think the research is particularly novel, but the identification of domains from the specified PICO criteria and the Delphi review of the identified domains is original and consequently, contributes original research evidence.

Authors: Thank you!

2/

Results reported have not been published elsewhere: Not for this study, of which I am aware.

Authors: no response required

3/

3. Experiments, statistics, and other analyses are performed to a high technical standard and are described in sufficient detail.

The scoping review may be sufficient for this form of research and appears to have been conducted in line with PRISMA-Scr guidelines. It would have been preferable for the grey literature to be assessed using the PICO developed for the academic database search. The grey literature contained a higher number of the relevant publications and yet, based on my reading of the text, the review process was separate from the specified PICO and accordingly introduced additional subjectivity, did not align with the PICO process/outcomes, and lost the focus on the research quality and impact assessment domains applied to research performance assessment/promotion. The argument could be made that the wider points drawn from the grey literature are pertinent, but if this was the case, the academic literature should have also drawn this wider remit. As a consequence, the scoping review conflates an historical review of the domains applied in the performance review/assessment process and a broader review of what could be incorporated. The latter component was however excluded from the academic database search and consequently misses a large volume of potentially relevant literature. Given the existing summation of potential domains provided the basis for the Delphi process, it is difficult to see how this could be comprehensively adjusted. Greater clarity regarding the inclusion/exclusion criteria for the grey literature may assist. This conflation is also relevant to the following point and the results, discussion and conclusion issue identified below.

The specified aim of the paper is 'to identify and prioritize DORA-compliant measures for assessing research.' This aim is inconsistent with the PICO, which, necessarily in this reviewer's subjective opinion, focuses on 'Personnel Selection/personnel management/employee incentive plans/employee performance appraisal/academic performance/career mobility' i.e. the assessment of researchers within a view to workplace performance appraisal, promotion, etc. I would recommend tightening the aim to reflect this narrower focus and similarly to refine the specified purpose to also reflect the performance context for the relevant measures.

Authors:

1/

To provide more detail about eligibility criteria, and by doing so dispel the concern about discrepancies in focus/eligibility between the database search and the grey literature search we edited and added details as follows. Overall, we searched both published research and grey literature for measures that can be used to assess the performance of researchers, and we applied the same eligibility criteria to both published research and grey literature. 

In Methods, Scoping Review, Eligibility (page 8, lines 150-151), we added: “All authors reviewed and refined PICO-based eligibility criteria, which applied to both the database search and the grey literature search.”

In S1 Appendix, we changed the title to “Appendix 1. Inclusion and exclusion criteria.” Under Publication Design/Type, we added grey literature. 

In Methods, Scoping Review, Grey Literature (page 9, lines 190-191), we added: “ARG searched for relevant reports that met eligibility criteria (S1 Appendix) and extracted data on…”

2/

We reframed the Aim (page 6, lines 111-113) to: “To achieve this, we aimed to identify and prioritize measures for assessing the performance of researchers that comply with DORA principles by focusing on quality and impact rather than metrics such as Journal Impact Factor.” We reframed the intent or application of the findings (page 6, lines 116-118) to: “…and academic organizations can employ these measures and related processes to assess the performance of researchers in the context of hiring, annual reviews, tenure, promotion and other decisions based on the quality and impact of research.” We reframed the Purpose statement (page 6, lines 119-120) to: “The purpose of this manuscript is to describe our methods and the resulting DORA-compliant measures for assessing the performance of researchers.”

4/

Conclusions are presented in an appropriate fashion and are supported by the data.

The paper results, discussion and conclusion needs to be carefully reviewed to improve precision and alignment with the PICO focus on researcher performance appraisal. For example, greater clarity is required to explain how Table 1/S6 Appendix links to the aims (or preferably narrower aims that reflect the PICO strategy).

Authors:

RESULTS

To help readers better understand what we mean by “processes”, and why we included S6 Appendix and Table 1, in Methods, Scoping Review, Data Collection and Analysis (page 9, lines 182-183), we added: “…ARG compiled a list of unique measures and related processes that could be implemented to promote awareness and support use of the measures reported or recommended across all included studies.”

In Results, Compiled Measures and Processes (page 13, lines 266-270), we added: “S6 Appendix shows the list of unique processes that organizations can implement to promote awareness and support use of those measures. Table 1 includes select illustrative examples of those processes, which create a culture conducive to DORA-compliant assessment of research performance based on research quality and impact.”

DISCUSSION 

The first paragraph briefly summarizes high-level findings, and the next two paragraphs situate our research in the context of prior research. We added changes to the next paragraph (page 20-21, lines 344-350) to emphasize the context of measuring the performance of researchers. 

CONCLUSION

We made similar edits in the concluding paragraph (page 24, lines 433-434) to emphasize the context of measuring the performance of researchers. 

5/

The article is presented in an intelligible fashion and is written in standard English: Yes

Authors: no response required

6/

The research meets all applicable standards for the ethics of experimentation and research integrity: Yes

Authors: no response required

7/

The article adheres to appropriate reporting guidelines and community standards for data availability: Yes

Authors: no response required

8/

Additional general points

As per the critique relating to the drift of focus from research performance assessment/promotion, the title is similarly ambiguous. I would recommend changing the title, aim, purpose, and where possible, the results, discussion and conclusion to this narrower focus. The field of research on research is generally under-researched, the initiative is valuable and the focus on researcher performance assessment is a highly relevant issue.

Authors: See changes described above to address concerns about the Aim, Purpose, Methods, Results, Discussion and Conclusion. We changed the title to: “DORA-compliant measures of research quality and impact to assess the performance of researchers in biomedical institutions: review of published research, international best practice and Delphi survey”

REVIEWER #2

1/

The review and Delphi study aimed to identify and prioritize DORA-compliant measures for assessing research quality and impact. Broadly speaking, this work is important and timely as we need better metrics for determining the quality and impact of research. This work has the potential to contribute greatly to the field. Overall, this manuscript is very high quality and meets standard reporting guidelines. It is transparent, thorough, and enjoyable to read. It is difficult to find areas for further improvement; however, I suggest two minor revisions:

Authors: Thank you!

2/

Please include search strategy for all databases, the appendix only includes MEDLINE

Authors: As requested, we included search strategies for all databases in S2 Appendix. 

3/

Line 192 you discuss how eight categories “inductively emerged.” Please describe in detail the methods you used for this inductive process (e.g. how the data were coded, use of thematic analysis, content analysis, if it was independently in duplicate or collaboratively, etc.)

Authors: 

Please see lines 194-200, which provide details of the inductive process:

“First, ARG perused all measures using content analysis to organize measures of similar theme in categories reflecting those themes; for example, measures that explicitly mentioned or pertained to team science, participation, co-production, registration or sharing of research or open publication were categorized under Team/Open Science; and measures that explicitly mentioned or pertained to peer-reviewed research funding, applications funded or grants were categorized under Funding. Members of the research team independently reviewed, discussed and agreed upon the categorization scheme.”

---

## [Decision Letter · Decision Letter 1]

7 Dec 2022

DORA-compliant measures of research quality and impact to assess the performance of researchers in biomedical institutions: review of published research, international best practice and Delphi survey

PONE-D-22-16889R1

Dear Dr. Gagliardi,

We’re pleased to inform you that your manuscript has been judged scientifically suitable for publication and will be formally accepted for publication once it meets all outstanding technical requirements.

Kind regards,

Niklas Bobrovitz

Academic Editor

PLOS ONE

Additional Editor Comments (optional):

Reviewers' comments:

Reviewer's Responses to Questions

**Comments to the Author**

1. If the authors have adequately addressed your comments raised in a previous round of review and you feel that this manuscript is now acceptable for publication, you may indicate that here to bypass the “Comments to the Author” section, enter your conflict of interest statement in the “Confidential to Editor” section, and submit your "Accept" recommendation.

Reviewer #1: All comments have been addressed

2. Is the manuscript technically sound, and do the data support the conclusions?

Reviewer #1: Partly

3. Has the statistical analysis been performed appropriately and rigorously? 

Reviewer #1: N/A

4. Have the authors made all data underlying the findings in their manuscript fully available?

Reviewer #1: Yes

5. Is the manuscript presented in an intelligible fashion and written in standard English?

Reviewer #1: Yes

6. Review Comments to the Author

Reviewer #1: I am very supportive of the emphasis and focus of the research presented in the publication. I am wary of scope creep with respect to the technical aspects of the reviewed documentation, particularly given the journal's, PLOS', focus on rigorous methods and techniques. However, given the journal editor has declared that they do not see a conceptual mismatch between the academic database data and the grey literature data, then I am happy to support publication. Good luck with your future research.

7. PLOS authors have the option to publish the peer review history of their article (what does this mean?). If published, this will include your full peer review and any attached files.

Reviewer #1: **Yes: **Simon Deeming

---

## [Editor Report · Acceptance letter]

16 Dec 2022

PONE-D-22-16889R1 

DORA-compliant measures of research quality and impact to assess the performance of researchers in biomedical institutions: review of published research, international best practice and Delphi survey 

Dear Dr. Gagliardi:

I'm pleased to inform you that your manuscript has been deemed suitable for publication in PLOS ONE. Congratulations! Your manuscript is now with our production department. 

Kind regards, 

on behalf of

Dr. Niklas Bobrovitz 

Academic Editor

PLOS ONE